

# Development and evaluation of customized software to automatically align macula and optic disc centered scanning laser ophthalmoscope fundus images

M. Elena Martinez-Perez[1], Franziska G. Rauscher[2,3,4], Pingping Zhao[5] and Tobias Elze[2,5]

[1] Instituto de Investigaciones en Matematicas Aplicadas y en Sistemas, Universidad Nacional Autonoma de Mexico, Ciudad de Mexico, Mexico
[2] Leipzig Research Centre for Civilization Diseases (LIFE), Universität Leipzig, Leipzig, Saxony, Germany
[3] Medical Informatics Center - Department of Medical Data Science, Universität Leipzig, Leipzig, Saxony, Germany
[4] Institute for Medical Informatics, Statistics, and Epidemiology, Universität Leipzig, Leipzig, Saxony, Germany
[5] Schepens Eye Research Institute, Harvard Medical School, Boston, MA, United States

Corresponding author
Franziska G. Rauscher,
franziska.rauscher@medizin.uni-leipzig.de

## ABSTRACT

In ophthalmology, the angle between the center of the optic nerve head and the center of sharpest vision (foveola) is a posterior fundus landmark parameter of the retina of the human eye. Together with the optic disc-fovea distance, it characterizes the position of the optic nerve head in relationship to the foveola.

The optic disc-fovea angle markedly influences the regional distribution of retinal layer thickness patterns, specifically the retinal nerve fiber layer thickness measured at the optic disc. Thus, the optic disc-fovea angle needs to be determined and routinely taken into account in morphological glaucoma diagnosis and in the assessment of structure-function relationship in optic nerve diseases.

However, despite the urgency of this information, currently the optic disc-fovea line and its angle are routinely not measured. Obtaining it post-measurement requires manual registration of the macula and optic disc optical coherence tomography (OCT) imaging data. OCT manufacturer-delivered software does not provide automated image registration. Therefore researchers are forced to manually perform the alignment over different scanning regions. To fill this gap, we provide two software packages which can be applied to routinely acquired clinical OCT data to automatically align macula and optic disc images.

In this work, we introduce and comparatively evaluate two separate software packages (BloodVesselReg and OCTFundusReg) to automatically align macula and optic disc centered OCT volume scans based on their respective scanning laser ophthalmoscope (SLO) fundus images. BloodVesselReg implements an image registration and mosaicing algorithm based on retinal blood vessels. OCTFundusReg optimizes a general-purpose image registration toolkit to operate on SLO images. Both methods were independently developed by different subgroups of authors of this study using a training dataset of 18,047 eyes from a population-based study. The methods were tested on a dataset of 3,570 eyes from glaucoma patients, with success/failure assessed by visual inspection and compared to failure reporting of the methods themselves. BloodVesselReg had a slightly higher accuracy (94.7%) than

OCTFundusReg (93.9%). Both methods together failed on only 1% of the eyes. BloodVesselReg reported 165 out of its 190 failures. OCTFundusReg provides a continuous failureAlert parameter which resulted in an area under the receiver operating characteristics curve (AUC) of 0.91 from a logistic regression model. When including the difference of fitting related parameters between the two methods, the AUC improved to 0.95. Both methods had success rates of over 90% when applied in isolation to a clinical testing dataset. When applying them together, the rate of at least one of the method succeeding was 99%. The methods are highly promising for applications under real-world clinical conditions and might help to facilitate disease detection and monitoring over time.

## INTRODUCTION

Optical coherence tomography (OCT) (*Huang et al., 1991*) is a frequently applied imaging technology in the field of ophthalmology to visualize the retina in three dimensions at a high spatial resolution. In clinical practice, OCT is routinely applied to detect and monitor retinal diseases and optic neuropathies, such as glaucoma (*Quigley, 2011*). Typically, ophthalmic OCT devices perform volume scans of the retina, with point-wise in-depth measurements (A scans) aligned in close proximity shaping an in-depth slice (B scan), with multiple parallel adjacent B scans arranged to shape a three-dimensional cube (volume scan). These volume scans cover a restricted area of the retina. Eye diseases like glaucoma are not limited to small retinal locations but can manifest at multiple retinal locations distributed over wide areas. Therefore, during a single patient visit, often several volume scans are performed at different retinal locations of the same eye.

For optic neuropathies like glaucoma, the contemporary routine OCT measurement procedure at many specialty services is to simultaneously obtain one macula centered and one optic nerve head (ONH) centered volume scan per eye. These two volume scans are measured independently and overlap depending on individual eye anatomy. To easier detect retinal defects and their spatial distribution, *e.g.*, retinal nerve fiber layer thinning across the retina, it is helpful for clinicians to automatically spatially align the two volume scans to cover a continuous area over the retina. Furthermore, the combination of the two volume scans enables identification of the relative positions of the fovea (typically only contained on the macula centered scan) and the optic disc (fully covered only on the ONH centered scan). The disc and fovea locations are important markers of individual eye anatomy, as the disc-fovea axis is related to individual trajectories of major nerve fiber bundles which, in turn, are relevant to detect and interpret individual nerve fiber damages related to glaucoma. The disc-fovea axis has been suggested as the standard normalization axis for the coordinate system to interpret retinal nerve fiber thickness measurements around the ONH.

OCT volume scans are three-dimensional, which complicates their spatial alignment. However, clinical ophthalmic OCT devices routinely obtain a scanning laser ophthalmoscopy (SLO) fundus image immediately before each OCT volume scan. The SLO image covers a larger retinal area than the volume scan, and the volume scan coordinates are chosen by the machine so that it is fully contained in the SLO area. As the volume scan coordinates within the SLO image are accessible, aligning the SLO images indirectly aligns the volume scans as well to a level which allows to compare machine generated layer thicknesses and, most of all, to determine the disc-fovea axis.

The Spectralis SD-OCT machine (Heidelberg Engineering GmbH, Heidelberg, Germany), on which our study is based, is one of the most frequently used ophthalmic OCT devices in the world. The manufacturer-delivered software, without commercial add-ons, as it is used in many ophthalmic practices, does not provide automated image registration. This currently requires researchers to perform the alignment over different scanning regions manually.

The vast majority of registration methods published, instead of being focused on creating a mosaic between images of the same category, are focused on multimodal registration, for instance between fundus images and SLO images (*Miri et al., 2016*; *Almasi et al., 2020*). A general review of the classification of OCT image registration methods has been made by *Pan & Chen (2023)*. Broadly speaking, OCT image registration methods can be divided into two large groups: 1) volumetric transformation-based methods which seek to maximize the similarity between voxels by comparing a template image to the subject image. These methods do not require a specific anatomical model, but require that both, the template and the subject images, can be obtained using the same imaging protocol; and 2) the image feature-based methods, that use several distinct anatomical features to determine the transformation parameters. These features are typically landmarks, such as blood vessels, curves, surfaces, or a combination of these. For a more comprehensive review the reader is referred to *Pan & Chen (2023)*.

Our current work introduces and comparatively evaluates two separate software packages to spatially and automatically align macula centered and ONH centered OCT volume scans based on their respective SLO fundus images, illustrated in Fig. 1. Both methods are feature-based registration methods, whose features are blood vessels and intensity blocks and whose transformations are based on affine and similarity transformations, respectively. The two packages, namely the MATLAB toolbox `BloodVesselReg` and the R library `OCTFundusReg`, have been independently developed based on data from the large population based LIFE-Adult Study and were evaluated on an independent dataset of clinical measurements from Massachusetts Eye and Ear (MEE) glaucoma service.

## DATA

For this work, two retrospective datasets were used, one of them for the development of the two methods (training data) and a second one for their evaluation (testing data). The data were available to the researchers in a fully de-identified manner, and the respective parts of the study were approved by the corresponding ethics committees, as detailed below.

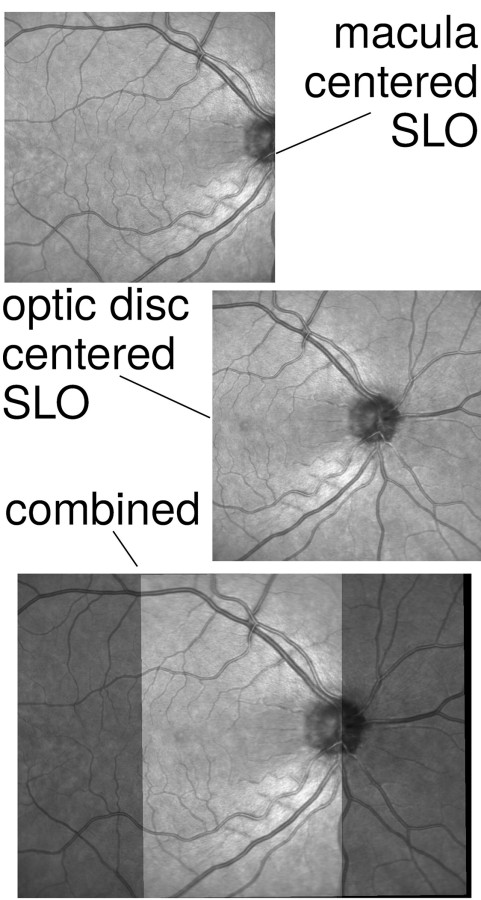

**Figure 1** Illustrative example of a macula centered (top) and an optic nerve head centered (center) scanning laser ophthalmoscopy (SLO) fundus image as well as their aligned combination (bottom), with the overlapping areas shown in lighter and the non-overlapping areas in darker gray.

## Training dataset

For the development of the two methods, Spectralis OCT (Heidelberg Engineering GmbH, Heidelberg, Germany) measurements from the population-based, age- and sex-stratified LIFE-Adult Study (*Loeffler et al., 2015*; *Engel et al., 2023*) were used. The LIFE-Adult Study baseline measurements, on which this work is based, were obtained between 2011 and 2014 from 10,000 randomly selected participants among the entire population of Leipzig, a city in eastern Germany with over half a million residents. The study was approved by the Ethical Committee at the Medical Faculty of Leipzig University (approval number: 263-2009-14122009) and adheres to the Declaration of Helsinki and all federal and state laws. Prior to inclusion, informed written consent was obtained from all participants. Recruitment was stratified by age and sex, with in total 400 participants in the age group from 19 to 39 years and 9,600 participants in the age group between 40 and 80 years. The majority of participants in the LIFE-Adult study are of European ancestry, with less than one percent having non-European ancestry.

## Data collection

During examinations, participants underwent extensive phenotyping, which included ophthalmological imaging, genetics, interviews, questionnaires, physical exams, and blood and urine tests. As part of the ophthalmic assessments, spectral domain optical coherence tomography (SD-OCT) imaging (Spectralis, Heidelberg Engineering) was used to obtain volume scans of the macular and optic nerve head regions, as well as cpRNFLT scans around the optic nerve head, following established protocols. The Spectralis SD-OCT incorporates an eye-tracking feature. For this, acquisition of B-Scan data is accompanied by simultaneous acquisition of one SLO image per B-Scan. The task of this feature is checking/avoiding saccades, as the OCT instrument performs checks during acquisition to see if one SLO is from the exact same location as the next SLO. This procedure is additionally necessary for follow-up images, where B-Scans are obtained within the location matching the SLO from the previous visit. For the present study, SLO fundus images were selected from all eyes with a complete pair of a macula centered and an ONH centered OCT scan. The images were transferred from the Spectralis device using the raw data export and converted to the TIFF format, keeping the original resolution of $768 \times 768$ pixels. In total, 18,047 SLO pairs from 18,047 eyes from 9,116 participants were selected as the training dataset for this study. The images were accessed for this research on 08.03.2023. Authors did not have access to information that could identify individual participants during or after data collection. In general, the eye imaging data is the basis of many analyses from our working group (*e.g.*, *Rauscher et al. (2024)*, *Baniasadi et al. (2020)*, *Li et al. (2020)*, *Wagner, Sommerer & Rauscher (2025a, 2025b)*, *Girbardt et al. (2021)*, *Rauscher et al. (2021)*, *Wang et al. (2017)*). Depending on the research question, cases with clinical eye disease are removed from the LIFE-Adult research dataset before analyses (*e.g.*, see respective flowcharts in *Baniasadi et al. (2020)*, *Li et al. (2020)*). However, in the current work we felt it was deliberately necessary to include cases with ocular disease within the dataset. This constitutes a strength of the current research, as our presented results are obtained even with eye disease present.

## Testing dataset

While an age and sex stratified population based study was used for the development of the two methods, we chose to evaluate the methods on a dataset from an eye hospital (MEE) to assess the applicability in clinical practice. The evaluation part of this work was covered by a protocol approved by the MEE Institutional Review Board (IRB).

Among all existing Spectralis OCT measurements from MEE glaucoma service patients scanned prior to 2021, the most recent pair of a macula centered and an ONH centered SLO image for each eye was selected. In total, for the testing dataset, 3,711 SLO pairs from 3,711 eyes from 2,108 patients were transferred from the machine using the raw data export and converted to TIFF format, keeping the original resolution of $768 \times 768$ pixels.

## METHODS

The spatial alignment of the partially overlapping SLO image pairs covering different parts of the retina (macula and ONH) requires an image processing technique called *mosaicing*.

We independently developed two separate mosaicing approaches for this task (`BloodVesselReg` and `OCTFundusReg`) which are described in detail in the next two sections. Afterwards, we describe the methodology of how these two methods were evaluated and compared. The source code of both methods is publicly available: https://github.com/tobiaselze/oct_fundus_registration. DOI: https://doi.org/10.5281/zenodo.13937462

## Mosaicing method 1: `BloodVesselReg`

`BloodVesselReg` has been implemented in MATLAB R2019b, using standard computer vision tools. As depicted in Fig. 2 the process consists of four main tasks: 1) preprocessing of images, 2) feature points extraction *via* blood vessel segmentation, 3) obtaining the matching points and the geometric transformation between images and 4) image projection *via* the homography matrix.

### Preprocessing

During the image acquisition process, it is possible to find sometimes a large contrast variation between images taken of different subjects, or even between images taken from the same subject at different times. Therefore the first step is to enhance the image contrast. The enhancement is applied only when the image needs to be enhanced, this decision is taken automatically based on the image histogram information.

When an image has poor contrast, its histogram is usually distributed in a very small dynamic range. To avoid enhancing all pairs of images in our dataset, we decided to use a general histogram statistics to determine a parameter that would give the threshold to take this decision. Preventing the possibility of having histograms with some degree of skew, we decided to use a robust technique to calculate the median of the 50% of the histogram data considering a non-normal distribution. The last median of square (LMedS) is a robust regression method typically used to eliminate outliers from a normal distribution (*Rousseeuw & Leroy, 1987*).

The least-squares technique (LS) consists of minimizing the sum of squares of the residuals. For linear univariate regression, these are given by $r_i = y_i - ax_i - b$, where $r_i$ is the residual of measurement $y$, and $a$ and $b$ are regression coefficients. A method with the theoretically highest breakdown point possible, namely 50%, has been proposed by *Rousseeuw & Leroy (1987)*, who replaced the least sum of squares by the least median of squares (LMedS). The LMedS estimator is given by Eq.(1).

$$LMedS = \text{minimize} \ \ med_i(y_i - ax_i - b)^2.$$ (1)

The LMedS estimator provides protection from outlying data, making it very appropriate for situations with errors. In our implementation the image contrast is enhanced if LMedS of the image histogram is $\leq$ threshold (taken to be $th_1 < 120$, almost half of the dynamic range).

When images need to be enhanced, an algorithm called *contrast-limited adaptive histogram equalization* (CLAHE) is used (*Pizer et al., 1987*). CLAHE algorithm overcomes the limitations of standard histogram equalization (HE) based on global image histogram

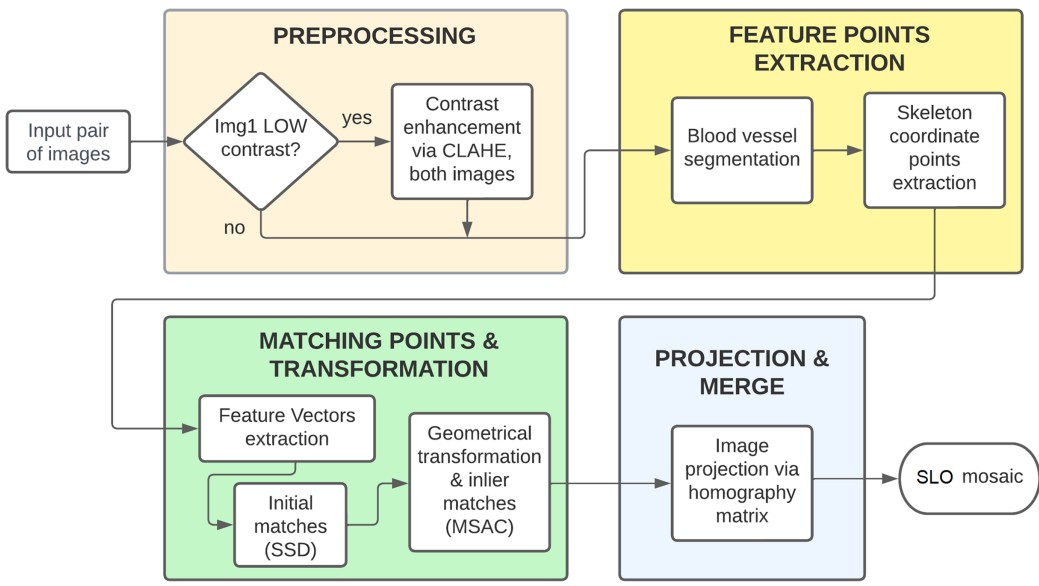

**Figure 2** Process for SLO mosaic construction with `BloodVesselReg` (customized software).

values. The two primary features of CLAHE are 1) adaptive HE (AHE), which divides the images into regions and performs local HE, and 2) the contrast limited AHE (CLAHE), which reduces noise by partially reducing the local HE. A bilinear interpolation is then used to avoid visibility of region boundaries. Figs. 3A and 3C show the original SLO images and Figs. 3B and 3D show the respective CLAHE enhanced images.

### Feature points extraction

Feature points are extracted from the center lines of the blood vessels.

- *Blood vessel segmentation and skeleton extraction:*
  A filter called *vesselness* was compute according to the method described by *Frangi et al. (1998)*. This function uses the eigenvectors of the Hessian to compute the likeliness of an image region to contain vessels or other image ridges. The Hessian matrix in 2D, of an image $I(x, y)$, is formed by the $2 \times 2$ matrix of second derivatives of $I(x, y)$ as shown in Eq. (2).

$$He = \begin{bmatrix} I_{xx} & I_{xy} \\ I_{yx} & I_{yy} \end{bmatrix} \tag{2}$$

Since $Ixy = Iyx$ the Hessian matrix is symmetrical with real eigenvalues and orthogonal eigenvectors which are rotation invariant. The eigenvalues of $He$ are ordered as $(|\lambda_1| \leq |\lambda_2|)$, where the following three quantities are defined in Eq. (3).

$$R_B = \frac{\lambda_1}{\lambda_2} \quad and \quad S = \sqrt{\sum_{j \leq 2} \lambda_j{}^2}; \tag{3}$$
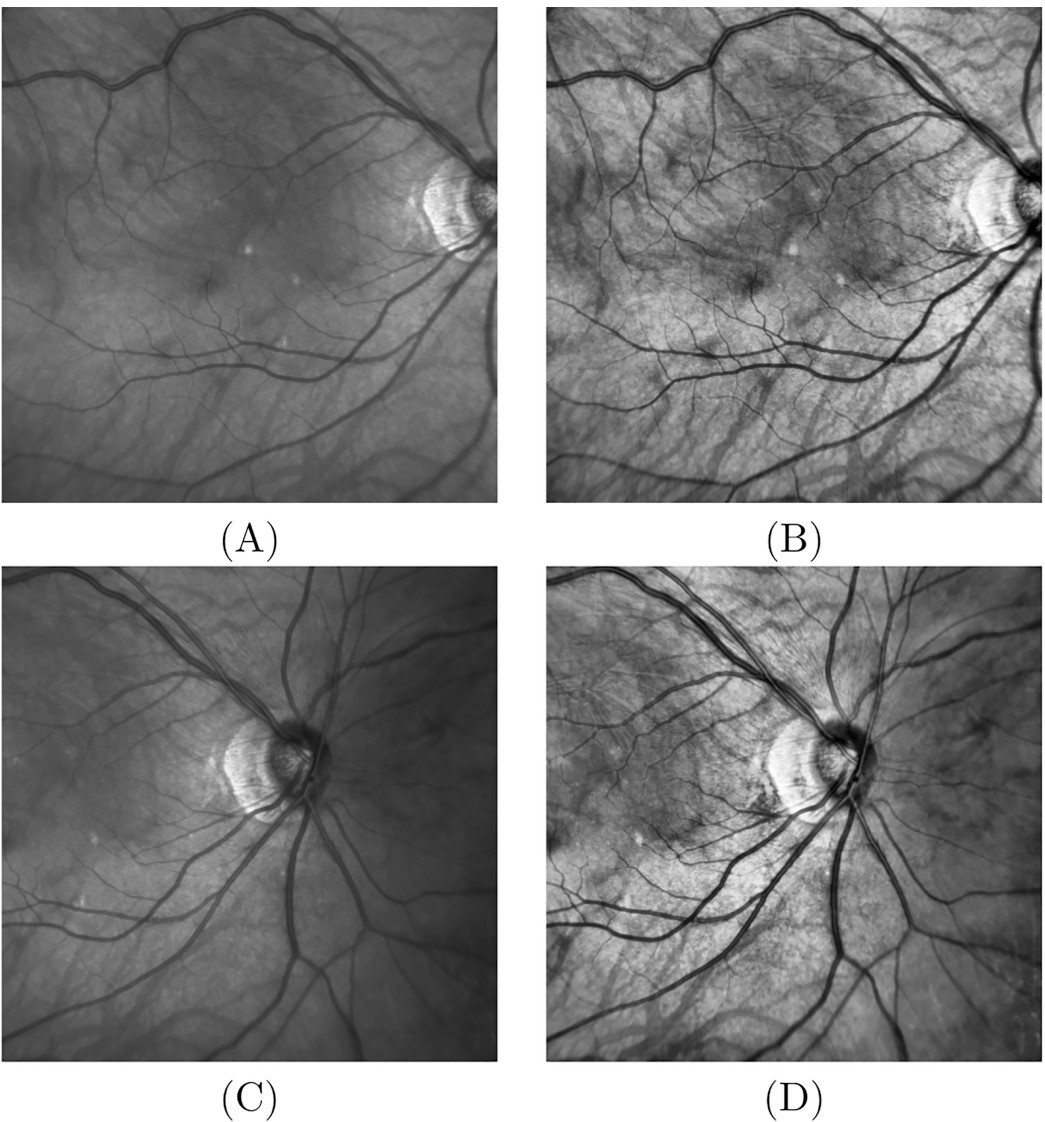

**Figure 3 Image enhancement.** (A) Original SLO image macula centered; (B) CLAHE enhancement of image (A); (C) original SLO image ONH centered; (D) CLAHE enhancement of (C).

$R_B$ is maximized for blob-like structures and should decrease as the structure becomes more vessel like, and $S$ is a measure of the relative brightness of the structure and should become large for vessels. Using the eigenvalues and Eq. (3), a vesselness measure is defined in Eq. (4) as:

$$V = \begin{cases} 0 & \text{if} \quad \lambda_2 > 0, \\ exp\left(-\frac{R_B^2}{2\beta^2}\right)\left(1 - exp\left(\frac{S^2}{2c^2}\right)\right) & otherwise \end{cases} \tag{4}$$

where $\beta$ and $c$ are normalizing values taken to be 0.5 and 0.5 max($S$) respectively. For the Hessian filter the second derivatives were calculated using second derivatives of Gaussian

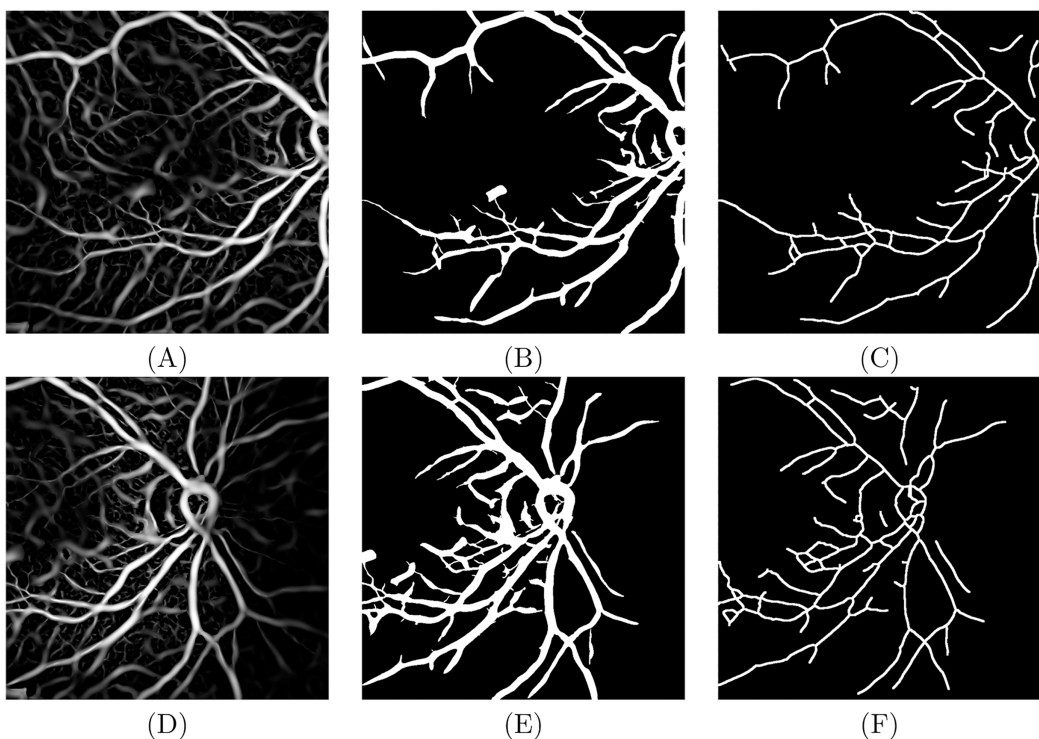

**Figure 4** **For image macula center: (A) Filter vesselness, (B) result of region growing and (C) a total of 5,405 vessel skeleton points.** For image ONH center: (D) Filter vesselness, (E) result of region growing and (F) a total of 6,352 vessel skeleton points.

functions of width $\sigma$, as the scale. At each pixel the maximum response over all the scales was selected. Based on previous experience an initial $\sigma$ of 0.8 was used. Figs. 4A and 4D show the results of applying the vesselness filter Eq. (4) to the image pair from Figs. 3B and 3D, respectively. Finally, the gradient magnitude of $V$, based on the first derivatives of $I$, $\nabla V = I_x + I_y$, is also computed for further analysis. Using these two features, $V$ and $\nabla V$, a region growing algorithm is used based on an iterative relaxation technique (*Martinez-Perez et al., 2007*). All the parameters used in the region growing are automatically calculated for each image from the histograms of the extracted features. The classification of pixels as vessel or background is based primarily upon the filtered image $V$, from which the criteria for determining seeds are defined. Using spatial information from the classification of the eight-neighbouring pixels, classes grow initially in regions with low gradient magnitude, $\nabla V$, allowing a relatively broad and fast classification while suppressing classification in the edge regions where the gradients are large. In a second stage, the classification constraint is relaxed and classes grow based solely upon $V$ to allow the definition of borders between regions. Figs. 4B and 4E show the results of the segmentation process. Finally, the vessel center lines are extracted *via* a thinning technique (*Lam, Seong-Whan & Ching, 1992*), see Figs. 4C and 4F.

### Computing the matching points and the geometrical transformation

The objective of the mosaicing is to register both images into a global coordinate system which contains the whole scene. Therefore one of the images will be set as a reference or simply the identity, and the other the one will be projected into the global coordinate system. The geometric transformation between the pair of planar images is obtained *via* the following homography: $p_r = H p_p$, where $p_r$ is the set of points in the reference image, and $p_p$ are those of the projected image. This expresion is written explicitly in Eq. (5).

$$\begin{bmatrix} x' \\ y' \\ 1 \end{bmatrix} = \begin{bmatrix} h_{11} & h_{12} & h_{13} \\ h_{21} & h_{22} & h_{23} \\ h_{31} & h_{32} & h_{33} \end{bmatrix} \begin{bmatrix} x \\ y \\ 1 \end{bmatrix} \tag{5}$$

where $p_r = [x', y', 1]$ and $p_p = [x, y, 1]$ in homogeneous coordinates. To estimate the geometrical transformation, a feature-based method that computes $H$ from a sparsely distributed set of point-to-point correspondences is used.

- *Extract feature vectors from the center line point positions:* For each of the image pair to be matched, the center line point positions (skeleton points) are used to extract feature vectors base on the intensity values of the filtered image, $V$. The block method was used for descriptor extraction, where every feature vector contains the intensity values of the $23 \times 23$ pixels block around the given point. Feature vectors are thus built as: $features = M \times 529$ and the corresponding locations are $validPoints = M \times 2$, where $M$ is the number of valid points, since the neighborhoods used are those that are fully contained within the image boundary, therefore the valid points may contain fewer points than the skeleton points. The skeleton points are shown in Figs. 4C and 4F, with 5,405 and 6,352 total skeleton points respectively.

- *Find the initial matching points from feature vectors:* An exhaustive method that computes pairwise euclidean distance between feature vectors is used. The feature vectors are first normalized to unit vectors before computation. The feature matching metric used was the sum of squared differences (SSD), where two feature vectors match when the distance between them is less than a threshold ($th_2 = 1.0$, empirically chosen as the default value). Fig. 5A shows the initial matches with 455 pairs of points, where the red points are in the reference image (I1, macula centered) and the green points are in the image to be projected (I2, ONH centered).

- *Find the inlier matches and the geometric transformation:* The outliers from the initial matches are excluded using the M-estimator SAmple Consensus (MSAC) algorithm (*Torr & Zisserman, 2000*). The MSAC algorithm is a variant of the Random Sample Consensus (RANSAC) algorithm (*Capel, 2004*). The more inliers, the better the model is. It does not matter how close the inliers actually are to the model, as long as they are within the threshold. MSAC uses the sum of all point-model distances as the quality measure ($th_3 = 4.5$ pixels), however outliers only add the threshold instead of their true distance. This method can lead to better results compared to RANSAC. Figure 5B shows the final inlier matches with 445 pairs of points.

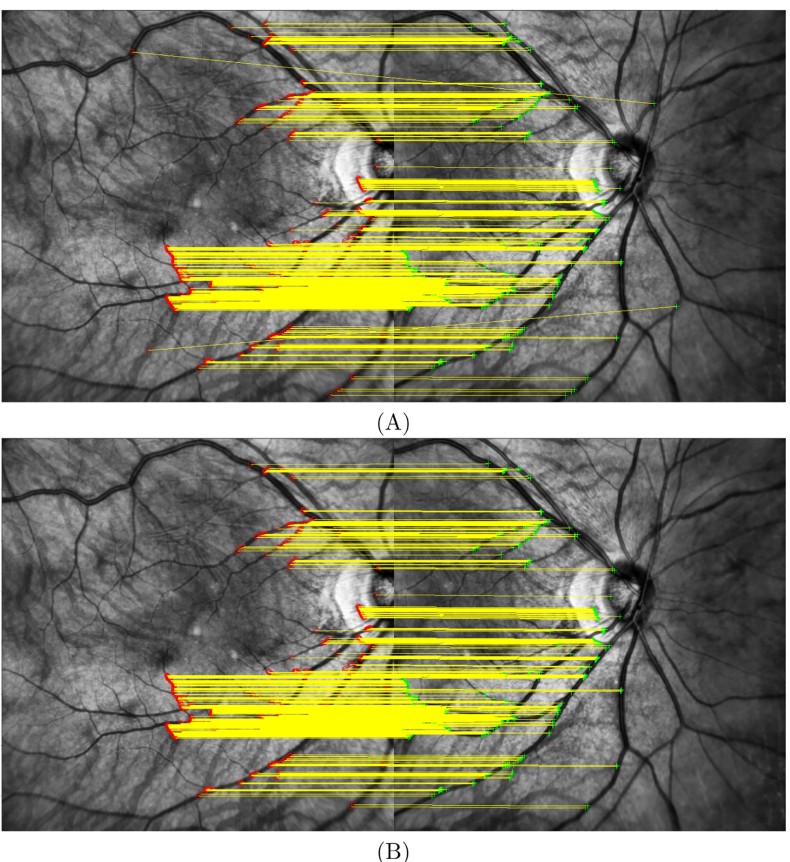

**Figure 5 Matched points.** (A) 455 initial matching point pairs and (B) 445 final inliers matching point pairs.

### Projection and merge

Transform all points to the same global coordinate system using $H$ (Eq. (5)). The whole mosaic image has now the reference coordinate of image I1 (macula centered). In order to display the points from I2 (ONH centered) into the same reference coordinate of image I1 (red points), the points of image I2 (green points) should be projected using the homography matrix, $p_r = H p_p$ Fig. 6.

### Accuracy of the projected points

Once the points of the I2 image are projected into the global coordinate system, the difference between the points of the image I1 (red) and those projected from the image I2 (green) should be close to zero.

Figure 7A shows both images before the mosaicing with the matched points in red for the reference image and in green otherwise. Figure 7B show the mosaic with all points from both images projected into the global coordinate system. Figure 7B has a total of 439 matched points.

The comparison between the values of reference *vs* projected points are as follow: average absolute difference in $x$ direction ($\Delta x$) $0.81 \pm 0.76$ pixels, average absolute difference in $y$ direction ($\Delta y$) $0.39 \pm 0.36$ pixels and average difference of euclidean
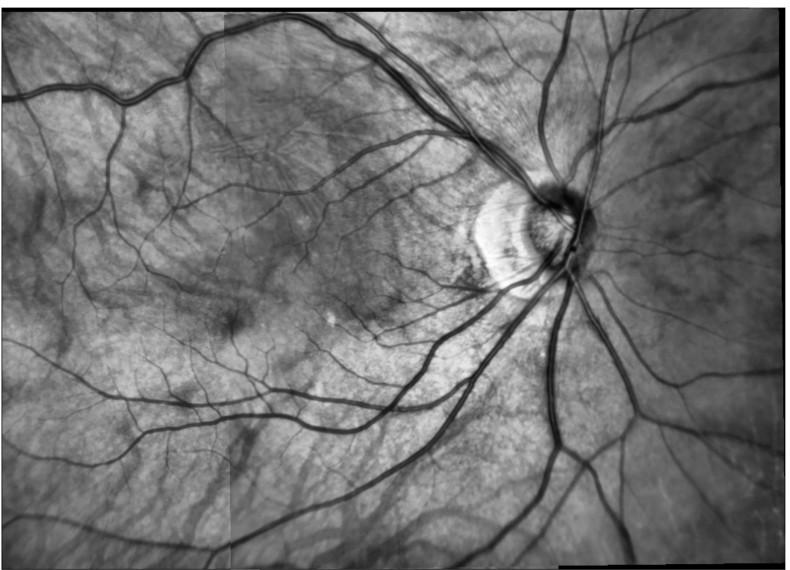

**Figure 6  Mosaic image.** Reference image centered in macula.

(A)

(B)

**Figure 7  Projection of points.** (A) Initial image macula centered (red points) and ONH centered (green points) and (B) projected points into the mosaic image coordinate system (reference system image center in macula).

**Table 1 The comparison between the values of reference *vs* projected points.** Average absolute difference in **x** direction ($\Delta x$), average absolute difference in *y* direction ($\Delta y$) and average difference of euclidean distance (D) between points. Average of the RMS error between reference and projected points, and average of the maximum RMS value. All metrics express in pixels, *n* = 250.

| Metric (pixels) | *mean $\pm$ std* (*n* = 250) |
|---|---|
| $\Delta x$ | $1.17 \pm 0.88$ |
| $\Delta y$ | $0.58 \pm 0.50$ |
| D | $1.42 \pm 0.88$ |
| RMS error | $0.88 \pm 0.80$ |
| Max (RMS) | 3.91 |

distance (D) between points $0.97 \pm 0.75$ pixels. In order to evaluate the accuracy of the projection the root mean square (RMS) error was computed. The RMS error measures the amount of misalignment between the reference points, $p_r$, and the projected points, $p_p$, as defined in Eq. (6) for *M* number of points:

$$RMS = \sqrt{\frac{1}{M}\sum_{i=1}^{M}\|p_{ri} - p_{pi}\|^2}. \tag{6}$$

The average RMS error for image in Fig. 7B is $0.60 \pm 0.63$ pixels and the maximum RMS value is 3.86 pixels. Table 1 shows a summary of all these metrics running on 250 mosaics from the training dataset.

**Mosaicing method 2:** `OCTFundusReg`

The second independently developed approach, `OCTFundusReg`, was implemented in R (*R Core Team, 2021*) by a sub-group of the authors of this study. Unlike `BloodVesselReg`, which is optimized for fundus images and aims to extract and to use fundus image specific blood vessel features for the mosaicing task, `OCTFundusReg` is based on an existing, open-source general-purpose medical image registration toolkit, NiftyReg (*Modat et al., 2014*), using its R interface RNiftyReg.

The default parameters of NiftyReg are optimized for the registration of fully overlapping images and did not work with the partially overlapping mosaicing task required for aligning the SLO fundus images. Therefore, as a starting value, the ONH centered SLO fundus image was shifted automatically 50% to the right of the macula centered SLO fundus image.

NiftyReg provides an internal parameter to assess the similarity of the images after registration. This parameter is based on normalized mutual information, with higher similarity values suggesting more successful registration results. Therefore, the similarity parameter seemed to be an appropriate candidate to assess the probability of successful registration. However, we noticed a dependency of this parameter on the area of overlap between the images, which, due to the substantial variety of overlaps depending on individual eye anatomy, defeated the purpose to use similarity alone as an indicator of

mosaicing success. Instead, the similarity parameter was included in a multivariable model to predict registration success, as detailed below.

As an initial step to create `OCTFundusReg`, we applied NiftyReg with affine transformation to the training dataset. Afterwards, we checked all aligned images (mosaics) by visual inspection for registration failures. To optimally predict registration successes and failures from the statistics of the affine transformations, we fitted a random forest model (*Breiman, 2001*; *Liaw & Wiener, 2002*) consisting of 500 trees with registration success as an outcome and the following covariates: NiftyReg's similarity (see above), plain normalized mutual information (NMI), percentage of overlap, horizontal transition, vertical transition, rotation, and skew.

After fitting this model, we applied a random forest variable selection model (VSURF; *Genuer, Poggi & Tuleau-Malot (2010*, *2015)*) to determine the truly relevant covariates. The random forest model consisting of those covariates was then included in our software to define the parameter *failureAlert* as 1 minus the ratio of the 500 decision trees predicting a registration success. Hence, *failureAlert* predicts the probability of a registration failure in our training dataset.

## Evaluation of the methods

Both methods were evaluated on a separate dataset not available during the generation of the methods. Unlike the population based measurements used to generate the methods, the testing dataset originates from clinical practice. All SLO fundus images from the testing dataset were visually checked for image quality and correctness. We allowed low quality images as long as both original images were good enough to perform the mosaicing test manually, otherwise they were excluded. Furthermore, image pairs were excluded if the two images of the pair did not belong to the same eye or did not contain the part of the retina specified by the respective label (*i.e.*, did not contain a macula centered and an ONH centered scan).

After applying both methods to the testing dataset, all resulting mosaics were visually inspected for registration failures. We compared the errors of both methods by a contingency table. In addition, we investigated how "self-aware" the two methods were with respect to their failures.

`BloodVesselReg` reports its own failure in cases where no matched points are found. We compared these self-reports to our visual inspections.

`OCTFundusReg` does not report registration errors directly but returns a parameter *failureAlert* (see above) bounded between 0 and 1 with higher values indicating higher probabilities of failures. To quantitatively assess this parameter, we fitted a logistic regression of *failureAlert* to the true failures detected by visual inspection.

Finally, we aim to improve the image registration task by combining both methods. If both methods are successful for an eye, the statistics of the resulting mosaic image should be highly similar. Discrepancies in the registration statistics, however, indicate that at least one of the two algorithms has failed. Comparing the statistics is complicated by the fact that `BloodVesselReg` returns a homography matrix, whereas `OCTFundusReg` results in

**Table 2 Comparison of mosaicing results between the two methods.**

| | | OCTFundusReg | | | |
|---|---|---|---|---|---|
| | | Success | Failure | Sum | Percent |
| BloodVesselReg | Success | 3,197 | 183 | 3,380 | 94.7% |
| | Failure | 154 | 36 | 190 | 5.3% |
| | Sum | 3,351 | 219 | 3,570 | 100% |
| | Percent | 93.9% | 6.1% | 100% | |

an affine transform. Both methods, however, allow an easy calculation of the percentage of overlap between the macula and the ONH centered SLO images after registration, calculated by the number of overlapping pixels relative to the total number of pixels of the source image. Therefore, we analyze the absolute difference in the percentage of overlap between the `BloodVesselReg` and the `OCTFundusReg` registration of the same eye.

## RESULTS

A total of 3,570 ONH-macula SLO fundus image pairs of 3,570 eyes of 2,035 patients passed the initial quality check by visual inspection and were included for testing the two methods. Table 2 compares the mosaicing successes and failures of the two methods. `BloodVesselReg` had a slightly higher accuracy on the testing set (94.7%) than `OCTFundusReg` (93.9%). Both methods together failed on only 36 out of the 3,570 eyes. All other failures were specific to only one of the two methods.

### Self-awareness of failures

`BloodVesselReg` reported 165 out of its 190 failures, which means it was unaware of 25 of its failures (0.7% of the total number of eyes). `OCTFundusReg` does not report failures directly but returns a continuous *failureAlert* parameter bounded between 0 (lowest failure probability) and 1 (highest failure probability). Figure 8A shows box plots of that variable *vs*. true RNiftyReg failures/successes. The quartiles of the two distributions (vertical borders of the boxes) are distinctly different. An according fitted logistic regression model has an area under the ROC curve (AUC) of 0.91 (Fig. 8B).

Among the 25 of its failures `BloodVesselReg` was unaware of, 21 succeeded (*failureAlert* quartiles: 0.000, 0.008, and 0.044) and four also failed (*failureAlert* quartiles: 0.189, 0.422, and 0.605) in `OCTFundusReg`.

### Combined outcome of the two methods

We define *overlapdiff* as the absolute difference in the percentage of overlap between the `BloodVesselReg` and the `OCTFundusReg` registration of the same eye. If we exclude those 165 images where the `BloodVesselReg` algorithm self-reported its failure and fit a logistic regression to the remaining images to predict (undetected) failures from *overlapdiff*, we receive an AUC of 0.95. For `OCTFundusReg`, if we include *overlapdiff* as a further regressor in addition to *failureAlert*, the AUC increases from 0.91 (*failureAlert* alone) to 0.96 (*failureAlert* + *overlapdiff*).

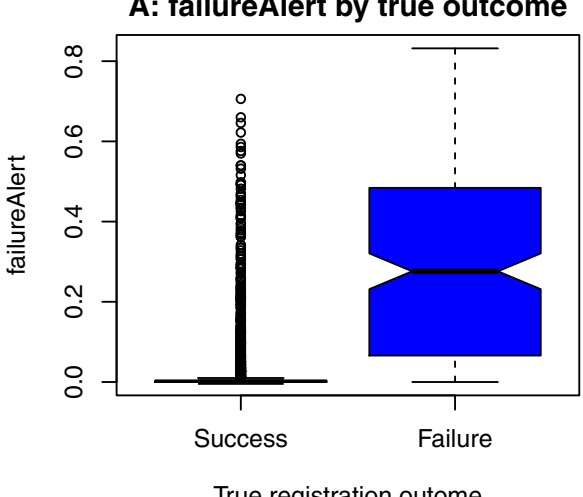

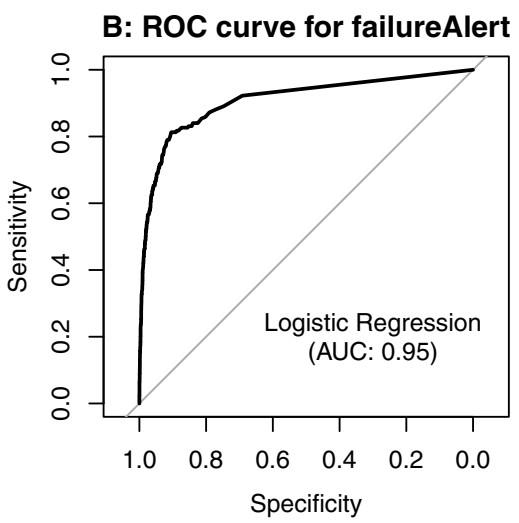

**Figure 8 Box plots of *failureAlert* by true successes and failures (A) and receiver operating characteristics (ROC) curve for a logistic regression model of *failureAlert* (B).**

## DISCUSSION

The standard measurement protocols of ophthalmic OCT machines scan limited parts of the retina at high spatial resolutions. As several high-prevalence eye diseases, such as glaucoma, may affect larger retinal areas than covered by these separate measurement protocols, several scans per eye and visit are performed in isolation, most frequently a measurement of the area around the macula and a second measurement of the area around the optic nerve head. Their spatial alignment to a larger, combined scan would provide clinicians with a better global picture of damages and might help to diagnose disease at an earlier stage or to more accurately monitor its progression over time. For this study, two research groups teamed up who had previously independently developed algorithms to perform this alignment. In this article, we introduce these two methods, publish their

source code, and test them independently as well as comparatively on the same clinical dataset.

## Two novel methods

Method `OCTFundusReg`, implemented as an R library, optimizes NiftyReg, an existing general-purpose image registration toolkit, for the specific task of mosaicing ONH and macula centered OCT generated fundus images. In addition to the affine registration parameters and the combined image, it returns a specific parameter to estimate the probability of a registration failure. It succeeded on 93.9% of the images of our clinical testing dataset.

Method `BloodVesselReg`, implemented in MATLAB, is an image registration and mosaicing algorithm specifically developed for fundus images. It identifies common features on the images based on blood vessels and uses these features subsequently for the alignment process. Unlike `OCTFundusReg`, it also allows non-affine transformations. It returns the homography matrix of the transform and the combined image. If no matching points are found on both images, it reports its own failure. It succeeded on 94.7% of the images of our testing dataset.

## Comparison of advantages and drawbacks of the two methods

Unlike `OCTFundusReg`, `BloodVesselReg` is tailored to the specifics of fundus images, as it extracts and exploits features based on retinal blood vessels. This makes the method robust and likely generalizable to other types of fundus images beyond the SLO technology. This general advantage, however, could under certain conditions be a shortcoming if the visibility of blood vessels is reduced within an image. In these rare cases, `OCTFundusReg` might have an advantage, as it was developed to work with all types of natural images and does not rely on blood vessels. Apart from that, the methods differ in their complexity: Unlike `OCTFundusReg`, `BloodVesselReg` allows non-affine transformations. A higher model complexity can be an advantage in cases of unusual image statistics, which might be difficult to fit with affine transformations, but it also poses a higher risk of overfitting.

## Development of evaluation metrics

The evaluation of the success of the registration proved to be unexpectedly complicated. In traditional image registration scenarios, where the overlap between the two images to be registered is close to 100%, the success of a registration procedure can typically be quantified by established numerical measures. State-of-the-art registration evaluation metrics are based on mutual information (MI), which is robust, flexible, and not limited to linear dependence, like traditional correlation coefficients. NiftyReg's internal similarity parameter is an example of such an MI-based method. In addition, we also calculated plain normalized MI (NMI) as a second established evaluation parameter. While these evaluation parameters typically work well on images with large overlaps, the unusually small and considerably variable overlap in our scenario posed a severe problem for these metrics. To better understand this problem, imagine, for example, two images having small

black edges on opposite corners and a registration algorithm which would register the two images exactly on these black edges, ignoring any other parts of the images. As the evaluation metric is then only applied to the overlap, and the small overlap consists of only black pixels on both images, the metric would assume the images to be perfectly aligned, as all pixels in this small area are identical. A visual inspection would immediately render this mathematically perfect registration useless and classify it as a failure. Note that most likely, the statistics of the relative positions of the two images, such as horizontal and vertical shift, percentage of overlap *etc.*, would be "unusual" compared to a truly successful registration, but the MI based metric would of course be unaware of this.

In absence of a functioning metric to numerically quantify registration success, we decided to use evaluation by visual inspection as the "gold standard" outcome. Initially, we planned for a multi-staging system to rate the quality of the outcome, from "excellent" to "failed". However, it turned out that registration was either a full success, defined as indistinguishable from the best outcome the observer would assume to have achieved by manual registration, or a total failure, *i.e.*, obviously wrong and unusable for any purposes. Therefore, we only used binary ratings (success *vs.* failure). As noted above, the two state-of-the-art registration metrics we calculated were not satisfyingly associated with registration success. Therefore, we developed our own measure, *failureAlert*, by combining these two parameters with *all registration outcomes available from an affine transform* (percentage of overlap, horizontal transition, vertical transition, rotation, and skew). These seven parameters were used to fit a non-linear regression model (random forest regression), using the R library randomForest (v. 4.6-14) with the default settings for all hyperparameters. As several of these regressors might be redundant, we applied VSURF (*Genuer, Poggi & Tuleau-Malot, 2010*, *2015*), a popular variable selection procedure for random forest models. In short, in several consecutive steps, VSURF first removes entirely irrelevant parameters, regardless of inter-regressor correlations, and then refines the selection of the remaining variables by eliminating redundancies. In the optimal reduced model, the following regressors remained: NiftyReg similarity, normalized mutual information, percent overlap, horizontal transition, and skew.

## Evaluation results and implications

As explained above, the ground truth for our method evaluation was based on visual inspection of the combined ("mosaiced") images. In real-life scenarios, large numbers of existing OCT scans might need to be registered in clinical practice, and a subsequent visual inspection of all images for possible registration failures would be very time-consuming if not infeasible. Therefore, we also evaluated the "self-awareness" of registration failures of the two methods. For `BloodVesselReg`, 0.7% of its registration failures remained undetected by the method, *i.e.*, were not reported as failures. The probabilistic *failureAlert* parameter from `OCTFundusReg` provided an area under the ROC curve (AUC) of 0.91. When applying `OCTFundusReg`, the *failureAlert* parameter could be used to visually inspect images starting from the maximum *failureAlert* value, which would maximize the probability of failure detection if only a limited number of images could be visually inspected due to time constraints. Applying both methods independently to the same data

using the difference of the registration overlap as a comparison parameter resulted in an AUC of 0.95 and would therefore be the most efficient way to select images for visual inspection for errors.

## Limitations

Possible limitations of our approach include that the two software packages were currently only evaluated on one OCT machine, although this machine is one of the most frequently used ophthalmic OCT devices worldwide. A further potential limitation is that our mosaicing approach is based on 2-D projections rather than on three-dimensional volume scans. If the two volume scans to be registered would have been recorded at different axial angles, for example, it would not be guaranteed that the entire 3-D voxel space would be appropriately registered as well, even if the 2-D fundus images were aligned. The most frequent expected use cases for the alignment used in this study, however, are registrations of thickness maps and the determination of the disc-fovea axis. For both of these cases, the 2-D alignment achieved by our software packages would be sufficient. If a full 3-D alignment of two volume scans is really required, our 2-D alignment would still provide a useful starting condition for the more complicated full 3-D mosaicing task.

To sum up, the two image mosaicing methods introduced in this work had success rates of over 90% if applied in isolation to a clinical testing dataset. When applying both of them together, the rate of at least one of the method succeeding was 99%. Comparing the overlapping areas of both input images after mosaicing between both methods helps to efficiently detect registration failures even if unreported by the methods themselves. Our two methods introduced in this work are therefore highly promising for applications under real-world clinical conditions and might help to facilitate disease detection and monitoring over time.

## Future work

The majority of participants in the LIFE-Adult study are of European ancestry, with less than one percent having non-European ancestry. Future work should include investigations on melanin-related differences in retinal imaging, as melanin concentration in the retinal pigment epithelium (RPE) and choroid influences the appearance and interpretation of retinal fundus and scanning laser ophthalmoscopy (SLO) images. Understanding these optical density differences associated with melanin concentrations is critical for advancing diagnostic precision and developing imaging techniques that are robust across diverse populations. High melanin concentrations absorb a broad spectrum of light, particularly in shorter wavelengths. This alters the contrast and brightness of fundus and SLO images. Future studies should aim to quantify these effects across different wavelength and different imaging modalities (*e.g.*, autofluorescence, angiography).

## CONCLUSION

In this work, we introduced and comparatively evaluated two separate software packages to spatially and automatically align macula centered and optic disc optical coherence tomography volume scans based on their respective scanning laser ophthalmoscope

fundus images. This aids detection and interpretation of clinical findings. `BloodVesselReg`, implemented in MATLAB, is an image registration and mosaicing algorithm specifically developed for fundus images; and `OCTFundusReg`, implemented as an R library, optimizes NiftyReg, an existing general-purpose image registration toolkit. We make both source codes publicly available. The software packages presented enable generation of comprehensive retinal layer thickness maps. As the volume scan coordinates within the SLO image are accessible, aligning the SLO images indirectly aligns the volume scans as well-to a level which allows to compare machine generated layer thicknesses and, most of all, to determine the disc-fovea axis.

Since the SLO image mosaic contains both ROIs, disc and fovea, in the same coordinate system, future work will be focused on their automatic localization and calculation of the disk-fovea axis which is related, as mentioned above, to the individual trajectories of major nerve fiber bundles which, in turn, are relevant for detecting and interpreting individual glaucoma-related nerve damage.

## ACKNOWLEDGEMENTS

The authors thank all participants for their time committed to the LIFE-Adult study. Furthermore, the authors gratefully acknowledge Dr. Kerstin Wirkner (Leipzig, Germany) and the LIFE-Adult study team for their commitment to the eye investigation and corresponding examinations to make this analysis possible.

### Funding

This research was supported by LIFE Leipzig Research Center for Civilization Diseases, Leipzig University (LIFE is funded by the EU, the European Social Fund, the European Regional Development Fund, and Free State Saxony's excellence initiative; project numbers: 713-241202, 14505/2470, 14575/2470). Further support was received by Tobias Elze from Lions Foundation; Grimshaw-Gudewicz Foundation; Research to Prevent Blindness; BrightFocus Foundation; Alice Adler Fellowship; NIH K99EY028631 to Mengyu Wang; NIH R01EY030575 to Tobias Elze; NEI Core Grant P30EYE003790. Support was also given by the German Research Foundation (grant number DFG 497989466) to Franziska G Rauscher. The authors received support for the Article Processing Charge by the Open Access Publication Fund of Leipzig University, Germany. The funders had no role in study design, data collection and analysis, decision to publish, or preparation of the manuscript.

### Grant Disclosures

The following grant information was disclosed by the authors:
The EU, the European Social Fund, the European Regional Development Fund, and Free State Saxony's Excellence Initiative: 713-241202, 14505/2470, and 14575/2470.

Lions Foundation; Grimshaw-Gudewicz Foundation; Research to Prevent Blindness; BrightFocus Foundation; Alice Adler Fellowship: NIH K99EY028631 and NIH R01EY030575.
NEI Core Grant: P30EYE003790.
German Research Foundation: DFG 497989466.
Leipzig University, Germany.

## Competing Interests

The authors declare that they have no competing interests.

## Author Contributions

- M. Elena Martinez-Perez conceived and designed the experiments, performed the computation work, performed the experiments, prepared figures and tables, authored and reviewed drafts of the article, and approved the final draft.
- Franziska G. Rauscher conceived and designed the experiments, analyzed the data, authored and reviewed drafts of the article, responsible for eye exam in population-based study to obtain OCT data, and approved the final draft.
- Pingping Zhan performed the experiments, reviewed drafts of the article, and approved the final draft.
- Tobias Elze conceived and designed the experiments, performed the computation work, performed the experiments, prepared figures and tables, authored and reviewed drafts of the article, and approved the final draft.

## Ethics

The following information was supplied relating to ethical approvals (*i.e.*, approving body and any reference numbers):

The study was approved by the Ethical Committee at the Medical Faculty of Leipzig University (approval number: 263-2009-14122009) and adheres to the Declaration of Helsinki and all federal and state laws. Prior to inclusion, informed written consent was obtained from all participants.

## Data Availability

The source code of both methods is available at GitHub and Zenodo:

- https://github.com/tobiaselze/oct_fundus_registration.

- Purple-Skittles, & tobiaselze. (2024). tobiaselze/oct_fundus_registration: OCT Fundus Registration (oct_fundus_registration). Zenodo. https://doi.org/10.5281/zenodo.13937462.

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
