# Peer review of "Development and evaluation of customized software to automatically align macula and optic disc centered scanning laser ophthalmoscope fundus images"

_PeerJ Computer Science, doi:10.7717/peerj-cs.2621_

## Round 0.1 · original submission · Major Revisions

Dear authors,

Thank you for submitting your article. Reviewers have now commented on your article and suggest major revisions. We do encourage you to address the concerns and criticisms of the reviewers and resubmit your article once you have updated it accordingly. When submitting the revised version of your article, it will be better to address the following:

1. Validation of the findings should be presented with more appropriate metrics for the quality of the article.
2. More information about the dataset used should be provided.
3. Equations should be used with correct equation number. Please do not use “as follows”, “given as”, etc. Explanation of the equations should also be checked. All variables should be written in italic as in the equations. Their definitions and boundaries should be defined. Necessary references should be provided.
4. Pros and cons of the method should be clarified. What are the limitation(s) methodology(ies) adopted in this work? Please indicate practical advantages, and discuss research limitations.
5. Future research directions should be included.

Best wishes,

Reviewer 1 ·

Basic reporting

1. The introduction of the paper does not adequately mention previous work on aligning different scans, which is a crucial aspect to understand the advancements and contributions of this study. It is recommended that the authors provide a more comprehensive literature review on prior methods used for the alignment of OCT scans or similar imaging modalities.

Experimental design

1. For Section Mosaicing Method 1: BloodVesselReg, the description of the BloodVesselReg method lacks a detailed explanation regarding the selection of the predefined threshold value for initial matching points. The paper states that a threshold of 1.0 is used for determining initial matches based on the sum of squared differences (SSD) between feature vectors. However, there is no detailed justification or rationale provided for choosing this specific value. It would be beneficial for the authors to explain how this threshold was determined, whether it was empirically chosen or based on prior research, and how sensitive the results are to changes in this threshold.

2. For Section Mosaicing method 2: OCTFundusReg, the authors mentioned ‘The second independently developed approach, OCTFundusReg, was implemented in RR Core Team’. It is unclear whether OCTFundusReg is a method proposed by the authors or one developed in prior research. If OCTFundusReg was developed by others, the authors should treat it as a comparable method with BloodVesselReg and clarify its origin. If OCTFundusReg was proposed in this paper, the authors should explicitly state this and ensure that its development and novelty are clearly delineated.

3. For the OCTFundusReg method, the paper briefly mentions that a random forest model with 500 trees was used to predict registration success, based on several covariates. However, there is a lack of detailed information on why these specific covariates were chosen, how they influence the model, and any hyperparameter tuning that was performed. The authors should provide a more comprehensive explanation of the random forest model, including the rationale for selecting specific covariates, details on hyperparameter tuning, and the impact of these choices on the model’s performance.

Validity of the findings

1. For Section Combined Outcome of the Two Methods, it mentions that BloodVesselReg was unaware of 25 of its failures. However, it does not provide a detailed comparison of how these specific failures were handled by OCTFundusReg, particularly regarding the failureAlert parameter. It is unclear whether the failureAlert values for these 25 failures were low (indicating undetected failures) or high enough to identify the issues.

Reviewer 2 ·

Basic reporting

Please see attached PDF.

Experimental design

Please see attached PDF.

Validity of the findings

Please see attached PDF.

Additional comments

Please see attached PDF.

Annotated reviews are not available for download in order to protect the identity of reviewers who chose to remain anonymous.

---

## Round 0.2 · Minor Revisions

Dear Authors,,

We acknowledge receipt of your revised paper and thank you for your efforts. However, we regret to inform you that one of the previous reviewers did not respond to our invitation to review the revision. Furthermore, according to one reviewer, it is still not recommended that your article be published in its current format. Consequently, we advise you to revise the paper in light of the reviewers' comments and concerns before resubmitting it.

Best wishes,

Reviewer 2 ·

Basic reporting

No comment.

Experimental design

No comment

Validity of the findings

No comment

Additional comments

The authors have made significant changes throughout that have improved the manuscript. As the authors note, the ethnicity of the individuals is not known but likely skewed toward white individuals. Given how different SLO/fundus images appear with and without high melanin concentrations, it would be useful to include this caveat in the limitations or discussion sections. Because the ethnicities were not known, it can even be framed as future work further improving upon the Editor's 5th comment. Additionally, the methods should explicitly state that there was no exclusions of eye disease in the used datasets.

---

## Round 0.3 · accepted · Accept

Dear Author,

The invitation to review the revised manuscript was not responded to by the previous reviewer. I have assessed the revision myself and, in my view, your paper is now sufficiently improved following the last revision. It is therefore ready for publication.

Best wishes,